# Spatiotemporal structure of SARS-CoV-2 mutational frequencies in wastewater samples from Ontario

**Paula Magbor** [1], **William Z. Wang** [1], **Gopi Gugan** [1], **Abayomi S. Olabode** [1], **Devan G. Becker** [2], **Valeria R. Parreira** [3], **Opeyemi U. Lawal** [3,4], **Amber Fedynak** [3], **Linkang Zhang** [3], **Fozia Rizvi** [3], **Melinda Precious** [3], **Christopher T. DeGroot** [5], **Lawrence Goodridge** [3], **Art F. Y. Poon** [1]*

1 Department of Pathology and Laboratory Medicine, Western University, London, Canada, 2 Department of Mathematics, Wilfrid Laurier University, Waterloo, Canada, 3 Canadian Research Institute for Food Safety, Department of Food Science, University of Guelph, Guelph, Canada, 4 School of the Environment, University of Windsor, Windsor, Canada, 5 Department of Mechanical and Materials Engineering, Western University, London, Canada

☯ These authors contributed equally to this work.
* apoon42@uwo.ca

**Data availability statement:** Study data have been deposited in a public repository in Zenodo at https://doi.org/10.5281/zenodo.15269976.

## Abstract

Starting October 2021, the Ontario wastewater surveillance initiative has used next-generation sequencing (NGS) to monitor SARS-CoV-2 RNA in wastewater samples. The fragmented and heterogeneous nature of these data precludes using comparative methods that require full-length genome sequences. In this study, we investigate the utility of the inner product of the vectors of mutation frequencies to quantify the temporal and spatial structure of these data. Raw sequence data were trimmed and mapped to the SARS-CoV-2 reference genome to extract mutation frequencies and coverage statistics. These data were filtered for samples with incomplete metadata, positions with insufficient coverage (>100 reads), or mutations with frequencies below 1%. For every pair of samples, we calculated the inner product of the respective mutation frequency vectors, and normalized the result to obtain a cosine distance. In total, we processed 1,619 samples from October 2021 to June 2023. The average depth was 7,693 reads, with mean coverage of 24,853 nt. A total of 241,078 mutations were detected in these samples. We restricted our analysis to 20 consecutive months with samples from at least one health region per month. A projection of the resulting cosine distance matrix revealed substantial temporal structure largely driven by the rapid spread of variants of concern. Genetic similarity, as quantified by the normalized dot product of mutation frequencies, was significantly negatively correlated with the geographic distance between sampling locations. These results suggest that spatial differentiation in the genomic variation of SARS-CoV-2 among wastewater samples can be measured, even at the relatively small scale of a single province.

**Funding:** This work was supported by funding from the Ontario Ministry of Environment Conservation and Parks (MECP) and by a grant from the Canadian Institutes of Health Research (PJT-183832) to AFYP.

**Competing interests:** The authors have declared that no competing interests exist.

## Introduction

Wastewater surveillance (WWS), also referred to as wastewater-based epidemiology, has a long history for infectious disease surveillance. One of the first examples of WWS was its use in 1939 to confirm the presence of poliovirus in sewage during an epidemic in South Carolina, USA [1]. The World Health Organization [2] adopted the use of WWS as part of its efforts for the global eradication of polio. Since then, this surveillance approach has been used to monitor other pathogens of public health importance, track antimicrobial resistance, and detect chemical and biological markers of pharmaceutical or illicit drug consumption [3]. WWS using standard molecular diagnostic tools such as PCR amplification and, more recently, next-generation sequencing (NGS) as a cost-effective, rapid, and non-invasive approach for disease surveillance [4]. It has presently become a significant component of modern epidemiology, facilitating the early detection and monitoring of emerging infectious diseases [5–7].

The COVID-19 pandemic highlighted the critical role of WWS as an effective tool for public health monitoring in real time, particularly for tracking the regional spread and evolution of SARS-CoV-2 [7]. In the fall of 2020, the provincial government of Ontario in Canada launched an Ontario Wastewater Surveillance Initiative (OWSI) led by the Ministry of Environment, Conservation and Parks [8,9]. One of the objectives of the OWSI was to use NGS to monitor the composition of SARS-CoV-2 genomic RNA in wastewater samples from treatment plants and other locations throughout the province. The OWSI then communicated information on trends in emergence and frequency of different variants of SARS-COV-2 transmission to public health agencies to inform their SARS-CoV-2 response efforts in Ontario [9]. At least 55 regions around the world, including the United Kingdom, the United States, France, the Netherlands, and Australia, have launched similar genomic wastewater surveillance programs [10,11]. These programs have played an important role in the detection of new outbreaks, the monitoring of emerging variants, and informing public health decision-making by governments and policymakers [12].

Using WWS to characterize SARS-CoV-2 genomic RNA presents several challenges. First, the NGS reads may result in insufficient or highly variable coverage of the virus genomes. Sequencing coverage can be influenced by variation in the efficiency of primer binding during the tiled amplification of viral RNA [13], which may be influenced by the emergence of a new lineage carrying mutations within primer-binding regions [14]. Low template concentrations in the nucleic acid extraction can be caused by fragmentation or degradation [15], or a period of relatively low community prevalence. A second challenge is that there is no information about the linkage between polymorphisms in reads derived from different amplicons. Wastewater samples containing SARS-CoV-2 nucleic acids will generally represent a large number of infections with limited divergence. Consequently, generating a single consensus genome from the sequences derived from a wastewater sample would represent a nonsensical average of a mixed population of infected hosts [16]. Reconstructing genomes of individual infections from such data is not feasible. Consequently, groups analyzing wastewater data have generally taken a different approach, assuming that the observed frequencies of mutations in a sample are drawn from a mixture distribution of latent (unobserved) components, where each component represents a different SARS-CoV-2 lineage, e.g., Freyja [17]. One must then estimate the relative contributions of each lineage to the mixture. While the lineage frequency estimates of these deconvolution methods are intuitive and interpretable outputs,

these estimates require detailed prior knowledge about the underlying lineages, i.e., the combinations of mutations that uniquely identify each lineage. These estimates also have a high level of uncertainty that is not consistently propagated to downstream analyses.

In this study, we explore a different approach to analyzing SARS-CoV-2 genomic wastewater data by focusing on the frequencies of individual mutations. This bypasses the computationally intensive step of mapping mutation frequencies to a reduced set of lineage frequencies. Our objective is to examine whether the genetic composition of wastewater samples varied significantly between geographic regions of the province at a given time, in addition to widespread changes over time driven by the spread of variants of concern. We hypothesize that the geographic distance between sampling sites leaves a detectable imprint on the distribution of mutation frequencies, despite the rapid transmission of SARS-CoV-2. Discernible spatial structure in the data would provide useful information on how the virus tends to spread throughout these regions, and inform public health strategies on tracking and responding to emerging variants.

## Materials and methods

### Study area

The OWSI focused primarily on two regions of Ontario: Southwestern Ontario and the Greater Toronto Area (GTA). Southwestern Ontario has a population of approximately 2.8 million and covers an area of about 37,000 square kilometers, bounded on the north by Lake Huron and to the south by Lake Erie. The GTA is the most densely populated region of Canada with a population of approximately 6.7 million, and covers an area of approximately 7,150 square kilometers, encompassing Toronto and surrounding municipalities. The high population density in these regions, combined with major international travel hubs and multiple land border crossings with the United States, makes the combined area ideal for using WWS to detect potential outbreaks and emerging variants of SARS-CoV-2.

### Data collection

A graphical outline of our analysis workflow is provided in S1 Fig. 24-h composite primary clarified sludge wastewater samples were collected and processed from multiple locations throughout southwestern Ontario (Fig 1), as previously described by [18]. Briefly, these samples were collected and concentrated using PEG precipitation. Next, RNA was extracted and sequenced at the University of Guelph using a tiled-amplicon approach, initially using the ARTIC V3 NCOV-2019 protocol (Oct to Dec 2021), and then transitioning through subsequent versions V4 (Dec 2021 to Feb 2022) and V4.1 (Mar 2022 to Aug 2023). We restricted our analysis to $n = 1,619$ samples collected between October 2021 and June 2023. In addition, we excluded $n = 20$ samples with incomplete metadata and potentially sensitive samples, e.g., samples associated with businesses and long-term care facilities. Metadata associated with samples were processed by harmonizing identifiers for site, municipality, geolocation and Ontario public health region. Irregular formats for sample collection dates were identified and standardized to conform to the ISO-8601 format. Samples with missing geolocation data were annotated based on the municipality they were collected in.

### Sequence analysis

Raw sequence reads were processed from the demultiplexed FASTQ files using cutadapt [19] to remove Illumina adapter sequences and discard reads below 10 nucleotides in length after

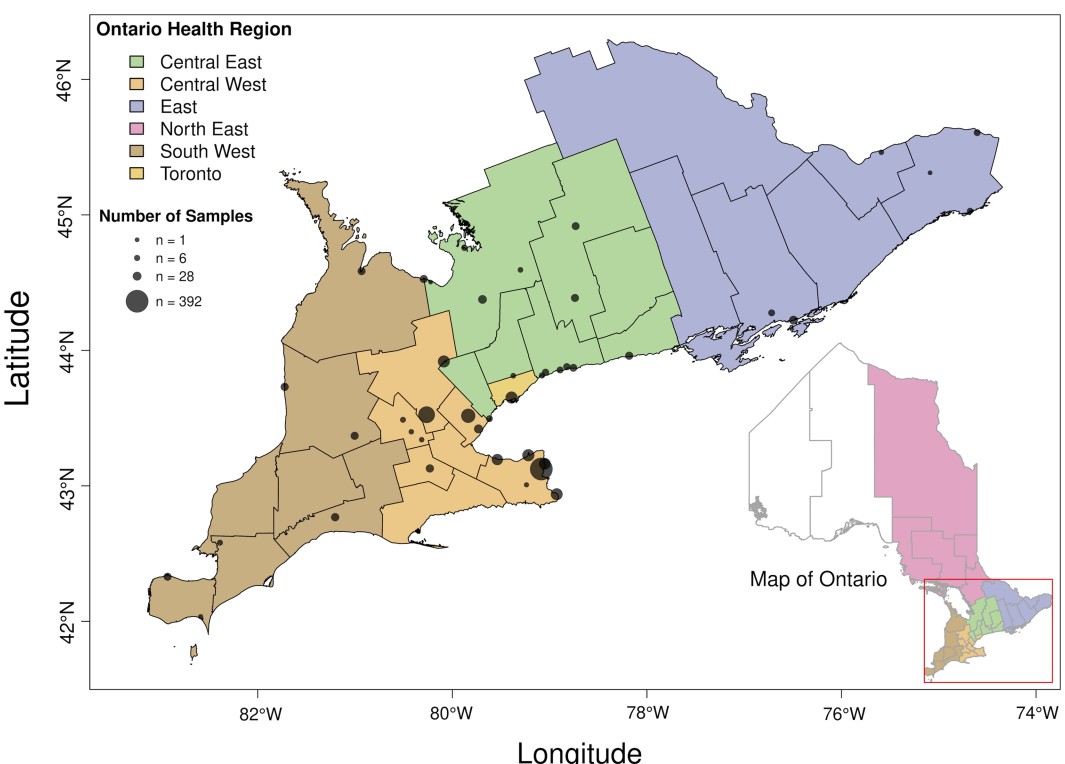

**Fig 1. Distribution of wastewater sampling sites in southern Ontario.** Municipalities are coloured by health region. In addition, *n* = 56 samples were collected in the northeast (not shown in figure). Circular markers indicate wastewater sampling sites using geolocation data when available or inferred from their associated municipalities. Marker size is proportional to the number of samples collected. Boundaries were reproduced from Ministry of Health Public Health Unit geospatial data under the Open Government License – Ontario, with permission from the King's Printer for Ontario, original copyright 2025.

trimming. We used minimap2 [20] to align reads to a reference genome (Genbank accession NC_045512.2), redirecting the standard output stream to Python to record the number of any mutations relative to the reference, and read coverage for each nucleotide position in the reference genome. These statistics were sufficient to calculate the relative frequency of every mutation in the sample. Any mutations with a coverage below 100 reads or a relative frequency below 1% were excluded from further analysis. The latter criterion was required to remove a substantial number of entries that were likely the result of sequencing error associated with Illumina platforms [21]. Python scripts used for these steps were released under the MIT license at https://github.com/PoonLab/gromstole.

We will use $f_x(i)$ to represent the relative frequency of mutation $i$ in sample $x$; if $f_x(i) < 0.01$ then we set $f_x(i) = 0$. Let $C(x)$ be the set of all mutations in sample $x$ with a minimum coverage of 100 reads. We calculated the inner (dot) product $\langle x, y \rangle$ as a similarity measure between samples $x$ and $y$:

$$\langle x, y \rangle = \sum_{i \in C(x) \cap C(y)} f_x(i) f_y(i) \tag{1}$$

where $C(x) \cap C(y)$ is the set of mutations with sufficient coverage in both samples. If this set intersection was empty, then we recorded $\langle x, y \rangle$ as a missing value for subsequent steps. We

normalized Eq 1 as follows:

$$\langle x,y \rangle' = \frac{\langle x,y \rangle}{\sqrt{\langle x,x \rangle \langle y,y \rangle}}. \tag{2}$$

which is equivalent to the cosine similarity that is often applied to sparse unstructured data [22]. Finally we converted the normalized dot product to a distance $D(x,y) = 1 - \langle x,y \rangle'$. Note that $\langle x,x \rangle$ and $\langle y,y \rangle$ were calculated over the same subset of mutations as $\langle x,y \rangle$, i.e., $C(x) \cap C(y)$. A useful property of $\langle x,y \rangle'$ is that it is scaled to the interval [0,1], where 1 indicates complete similarity. This normalization also conferred some robustness to variation in read coverage among samples. To understand this property, consider the set of frequencies in a sample, $f(x)$, as a vector in a high-dimensional space where every dimension represents a different mutation. The quantity $\langle x,y \rangle'$ represents the angle between two vectors rooted at the origin. Although a loss of coverage collapses some dimensions in this enormous space, the angle between vectors is largely preserved. Hence, cosine similarity has used as a basis for imputing missing data [23] and computing similarity matrices on incomplete data [24]. It is also a popular choice in the field of text mining due to its robustness to variation in document lengths [25].

## Statistical analysis and visualization

We applied both t-stochastic neighbor embedding (t-SNE [26]) and uniform manifold approximation and projection (UMAP [27]) methods in R to visualize the $D(x,y)$ distance matrix. Sample row/columns associated with 8 or more missing values in $D(x,y)$ due to poor coverage ($n = 11$) were excluded from the matrix. We used these dimensionality reduction methods to reduce the resulting matrix to two dimensions, which we used to visually screen for outliers and to assess whether there was evident temporal or geographical structure to the data. To adjust for variation in the number of samples over time, we randomly down-sampled the distance matrix to a maximum of 30 samples per time interval. These plots were also stratified by Ontario Health Regions (OHR) and time to investigate the spatial and temporal clustering associated with samples.

To assess the correlation between genetic similarity and geography, we used Pearson correlation tests between the distances $D(x,y)$ and the spatial distance (in kilometers) between sampling sites. Latitudinal and longitudinal coordinates were manually curated from sample metadata. When the metadata was incomplete, coordinates were inferred based on the locations of wastewater treatment plants listed on the respective municipality websites. We used a sliding window approach with a window size of four weeks and step size of two weeks to compute mean trends over time. These results were used to identify possible associations between geographic proximity and similarity in mutation profile. To quantify spatial clustering, we conducted permutation tests to compare the mean normalized dot product $\langle x,y \rangle'$ between samples $x$ and $y$ from the same Ontario health region (OHR) against pairs of samples from different OHRs. This yielded the following test statistic: $E[\langle x_j, y_j \rangle'] - E[\langle x_j, y_k \rangle']$ where $j$ denotes an sample's OHR, and $k$ denotes any OHR other than $j$. This calculation was performed using subsets of the data for each month between October 2021 and June 2023. If no samples were collected for a specific OHR in a given month, the test statistic was not calculated for that OHR. To generate a null distribution of the test statistic for each OHR, we randomly reassigned the OHR labels among all samples in the monthly subset. The test statistic was then recalculated for each reassignment, and this process was repeated 1,000 times.

## Results

### Data collection

We processed the raw NGS data derived from 1,599 wastewater samples collected between October 2021 and June 2023 across Ontario (Figs 1 and 2A), from which SARS-CoV-2 RNA was extracted, reverse-transcribed and amplified in 99 overlapping segments of about 400 nt each. The median number of paired-end reads per sample was 1,450,440 (interquartile range, IQR = 1,218,495 to 1,659,847; Fig 2B); only four (0.25%) samples comprised fewer than $10^5$ reads. The distribution of mapped reads was highly variable along the length of the reference genome (Fig 3) with median read depth ranging from 0 to 18,091 (median 3,330). A breakdown of read depth stratified by month of sample collection is also provided (S2 Fig). In particular, 2,711 (9.1%) nucleotide positions in the reference genome had a median read depth lower than 100, which was our threshold for calling mutations. The median number of positions with adequate read depth per sample was 27,019 (IQR 25,595–27,892) nucleotides. This uneven distribution of reads among samples highlights the significant challenge in using these data to analyze the composition of SARS-CoV-2 genomes in wastewater samples. One cannot rely on the sufficient coverage of mutations that would be necessary to resolve this variation down to the level of specific lineages of the PANGO nomenclature [28]. For this reason, we focused on the mutation as the fundamental unit of analysis rather than the lineage.

We recorded a median of 40,093 (IQR 28,324–48,079) mutations per sample, of which we discarded a median of 100 (IQR 71–139) mutations that occurred in positions with inadequate read depth (fewer than 100 reads). Furthermore, a median of 14,365 (39%; IQR 13,429–16,866) mutations per sample were observed in only a single read. Given that the error rate for Illumina sequencing is about 1% and that we required a minimum coverage of 100 reads, we would expect many of these mutations to be the result of sequencing error. Thus, we applied a conservative relative frequency threshold of 1% to all mutations, leaving a median of 283 (IQR 173–427) mutations per sample.

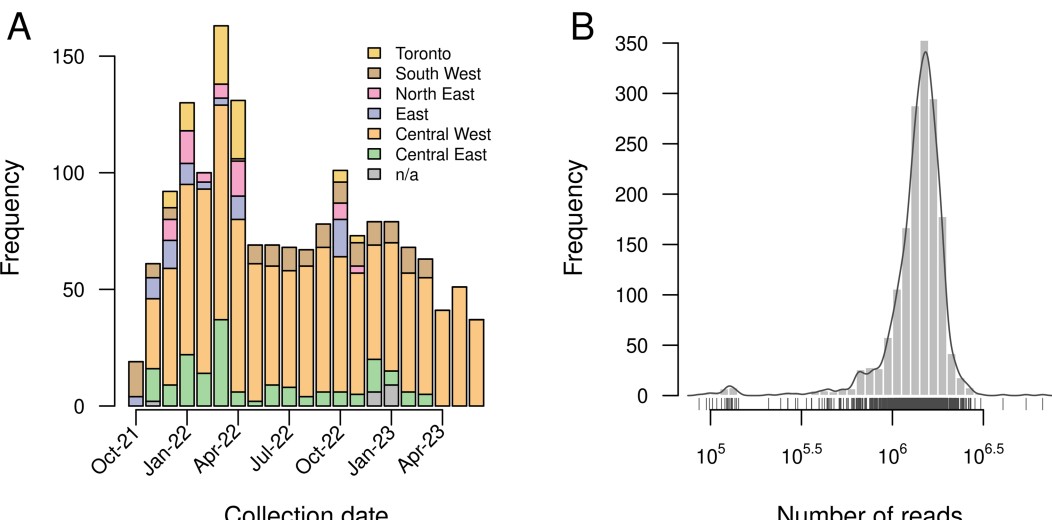

**Fig 2. Summary of sampling distribution and sequencing outputs. (A)** Numbers of samples collected per month, stratified by region. Region of sampling could not be determined for *n* = 17 samples due to incomplete metadata. **(B)** Distribution of the number of paired-end reads generated for each sample. We excluded one outlier sample with only 15 reads, which would otherwise cause the range in the number of reads to become too broad for visualizing the distribution.

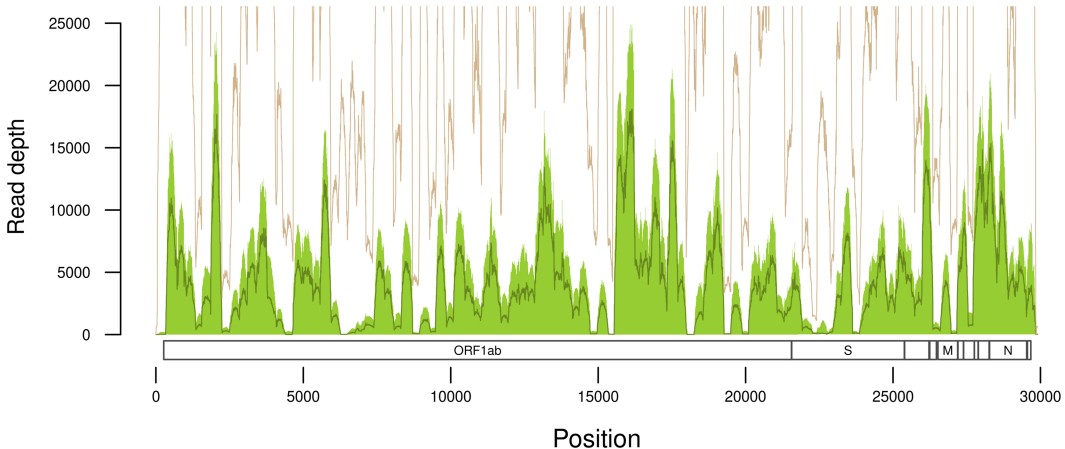

**Fig 3. Summary of read depth with respect to SARS-CoV-2 reference genome coordinates.** The shaded region and solid line within the region correspond to the interquartile range and median read depth, respectively. An additional solid line above the shaded region represents the maximum depth obtained for any sample (note this line frequently exceeds the upper limit of the plot region). The minimum read depth was zero for every position. Reference gene coordinates are displayed along the bottom of the plot.

## General sample composition

We recorded a total of 241,078 unique mutations across all samples. Of these, 73,872 unique mutations were observed at a minimum frequency of 1% and a depth of at least 100 reads in one or more samples. 48,055 (65%) of these mutations were nucleotide substitutions, of which 34,903 (72.6%) involved amino acid replacements. This proportion is significantly lower than what would be expected if mutations in the protein-coding regions of the SARS-CoV-2 genome occurred completely at random (77.2%, [29]), implying that some non-synonymous mutations were removed by purifying selection. In addition, 4,005 (5.4%) of the mutations were insertions, and 21,811 (29.5%) were deletions.

Deletions were not only more frequent overall than insertions, but they were also significantly longer on average (Fig 4). Specifically, insertions had a mean length of 1.1 (range 1–25) nucleotides, and deletions had a mean length 2.4 (range 1–92) nucleotides. We also observed that indels with a length divisible by three tended to be relatively more frequent for both insertions and deletions when compared to indels of a similar length (Fig 4), which is consistent with purifying selection removing indels that induce frameshifts within coding regions of the genome.

Most mutations appeared in only a few samples (Fig 5A). For instance, 67,034 (90.7%) of all mutations were observed in ten or fewer samples. The most prevalent mutation that was not fixed in all samples was orf1b:P314L, which was observed in 1,562 (96.5%) samples. Using negative binomial regression to adjust for overdispersion in the number of samples per mutation, we found that the average deletion was observed in significantly fewer samples than the other types of mutations (coefficient estimate $\beta = -0.94$, 95% CI = $-0.97$ to $-0.91$), where $\beta$ is the expected change in the log-transformed number of samples. In contrast, insertions tended to be found in significantly more samples ($\beta = 0.91$, 95% CI = 0.84–0.97), i.e., have a disproportionately broader distribution. This implies that while insertions less likely to occur *de novo*, they were also less likely than deletions to be subject to purifying selection. Averaging frequencies across samples in which each mutation was detected, we observed that most mutations were found at low frequencies overall (Fig 5B). For instance, 67,919 (91.9%) unique

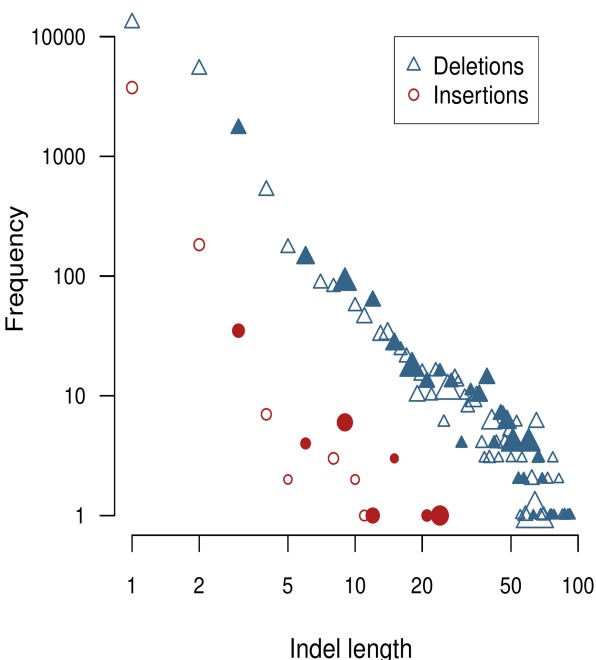

**Fig 4. Log-log scatterplot of the number of unique insertions (circles) and deletions (triangles) as a function of indel length in nucleotides.** Points are filled if the corresponding indel length is divisible by three. Point size is scaled in proportion to the average frequency across samples of indels in that length class.

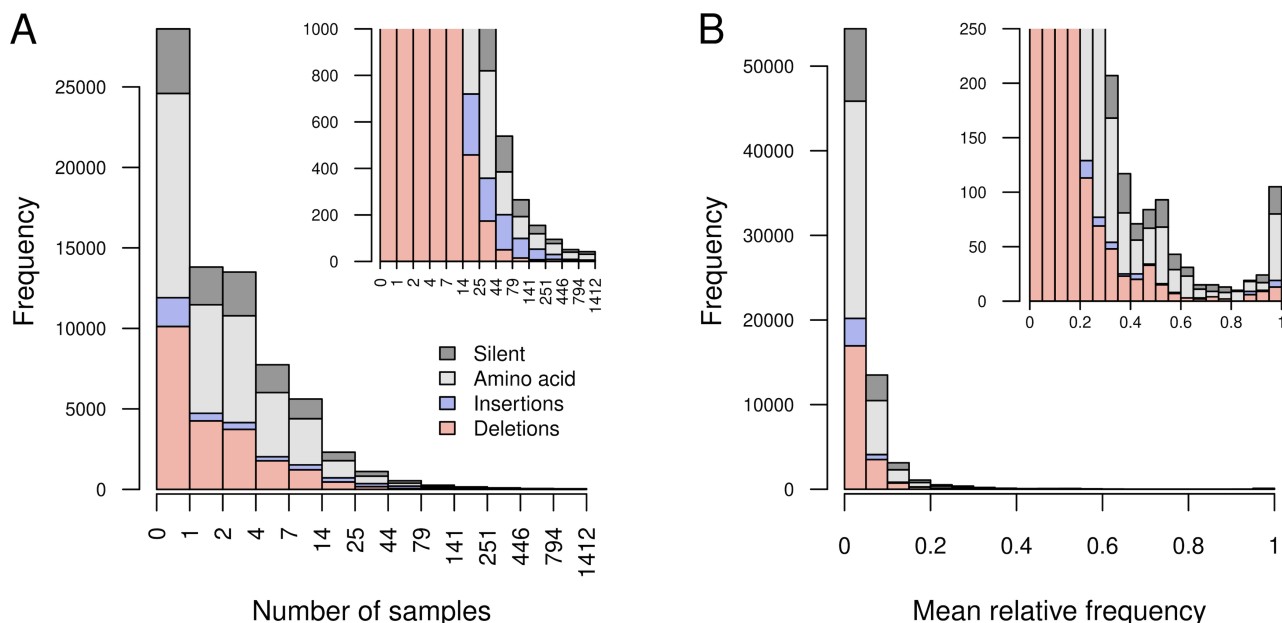

**Fig 5. SARS-CoV-2 mutation prevalence and mean relative frequencies across samples.** Stacked barplots summarizing the distributions of $n = 73,871$ SARS-CoV-2 mutations with respect to **(A)** the number of samples or **(B)** the mean frequency that mutations of each type (see legend) occurred at a minimum frequency of 1% and a minimum coverage of 100 reads. Labels on the $x$-axis for (A) denote the open-closed intervals for each set of bars, e.g., (4,7] contains 5, 6 and 7. Insets display the same plots rescaled to a narrow range on the $y$-axis.

mutations had mean frequencies between 0.01 and 0.1. Deletions and insertions were both observed at significantly lower frequencies than substitutions on average (Fisher's exact test, odds ratio OR = 0.62, 95% CI = 0.58–0.66).

## Mutational frequencies over time

A visual summary of the frequencies of mutations over time averaged across all samples and locations is provided as a heatmap (Fig 6). Because it was not feasible to visualize all 241,078 unique mutations recorded in these data, we limited this visualization to only 147 mutations that occurred at frequencies ranging from 5% to 95% when averaged across all samples. Mutations within this range tend to be driven to fixation by directional selection during the sampling period. These mutations included one insertion and 13 deletions; of the remaining 134 substitutions, 70.9% of them were non-synonymous (markedly less than expected by chance). The maximum frequency that a mutation was observed in any time interval was right-skewed with a median of 0.92 (IQR 0.77 to 0.98). In the majority of cases, mutation frequencies declined back toward zero as another lineage carrying the 'wildtype' at that position increased to fixation in the population. For instance, only 43 (29%) of these mutations were observed at a frequency above 90% in the last time interval (April 28 to May 26, 2023). Many of the mutations tended to have correlated frequencies due to their association with a variant of concern, e.g., BA.1. A disproportionately high number of mutations ($n$ = 56, 39%) in this subset occurred within the S gene. This proportion is significantly greater than expected by chance, given that the S gene is about 12.8% of the reference genome length (one-tailed binomial test, $P = 8.5 \times 10^{-15}$). In addition, only one of 51 substitutions affecting the S gene in this set of mutations was synonymous. These results are consistent with the tendency for strong selection to act on the spike protein.

## Dot product analysis

In order to compare the individual samples based on the frequencies of tens of thousands of mutations in the SARS-CoV-2 genome, we calculated the dot (inner) product for every pair of samples. Each sample was converted into a sparse, ordered vector of frequencies for every mutation ever recorded with sufficient coverage. This process yielded a matrix of angular distances. Starting from 1,599 samples, we discarded 11 samples with no overlap in coverage to one or more other samples, making it impossible to calculate the dot product. We first used multi-dimensional scaling (MDS) to examine the resulting distance matrix, which identified a single outlier. This outlier corresponded to a sample with low coverage (570 out of 29,903 reference nucleotides) and was discarded from further analysis. In addition, we observed that a majority of eigenvalues were negative, indicating that the distance matrix was highly non-Euclidean. Consequently we switched to non-linear dimensionality reduction methods to visualize the relationships among samples, namely t-stochastic neighbor embedding (t-SNE) and uniform manifold approximation and projection (UMAP).

Fig 7 depicts the results from t-SNE with samples labeled by month of sample collection. This projection clearly illustrates the temporal structure in mutation frequencies among samples; similar results were obtained using UMAP (S3 Fig). We observed some discontinuities in the distribution of samples in this projection, which are consistent with rapid fixation of variants of concern carrying substantial numbers of mutations, which was observed for Fig 6. However, it was difficult to distinguish differences among health regions in the context of the dominant temporal structure of this projection (S4 Fig). Differences between health regions were largely driven by variation in sampling rates over time (Fig 2).

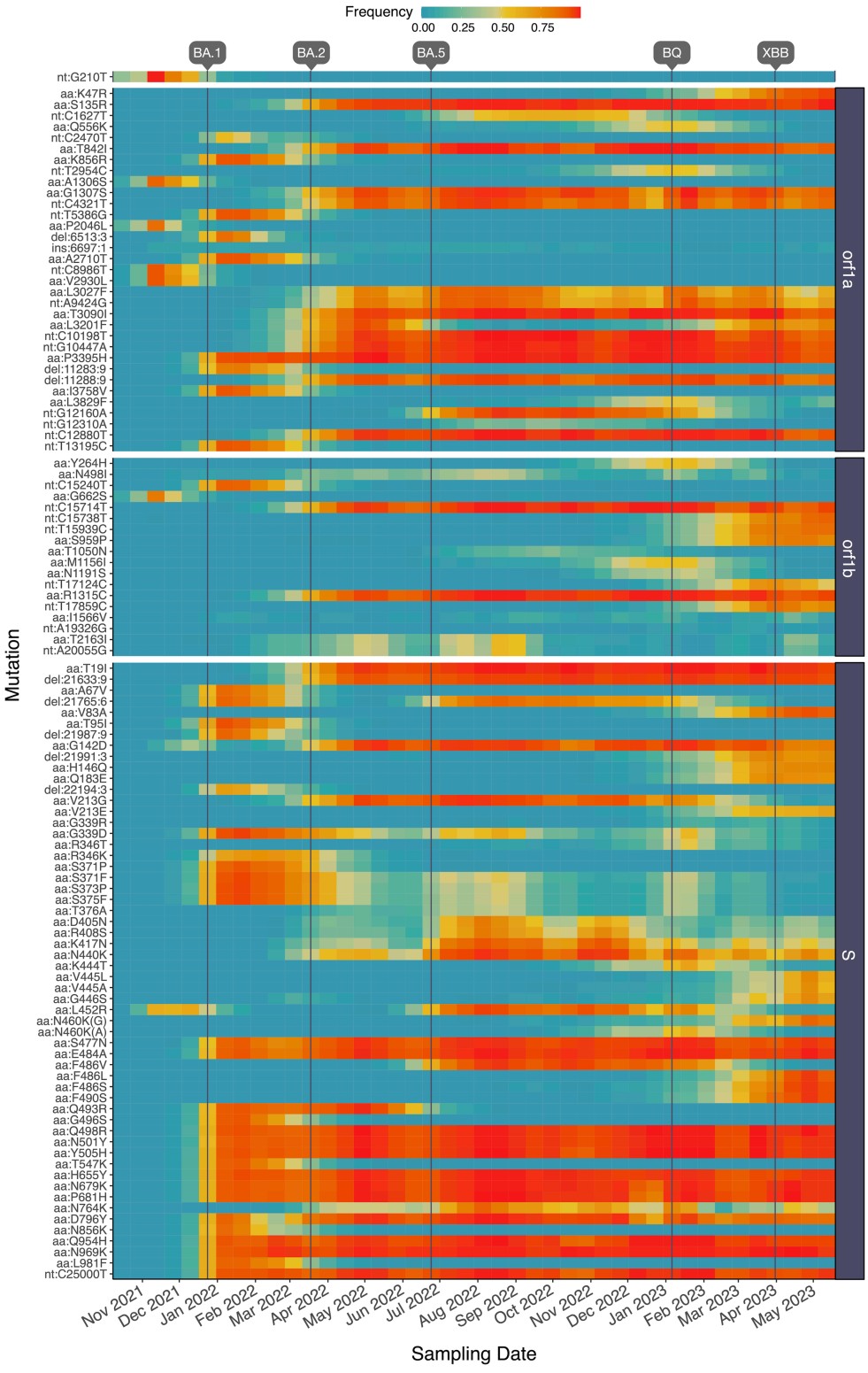

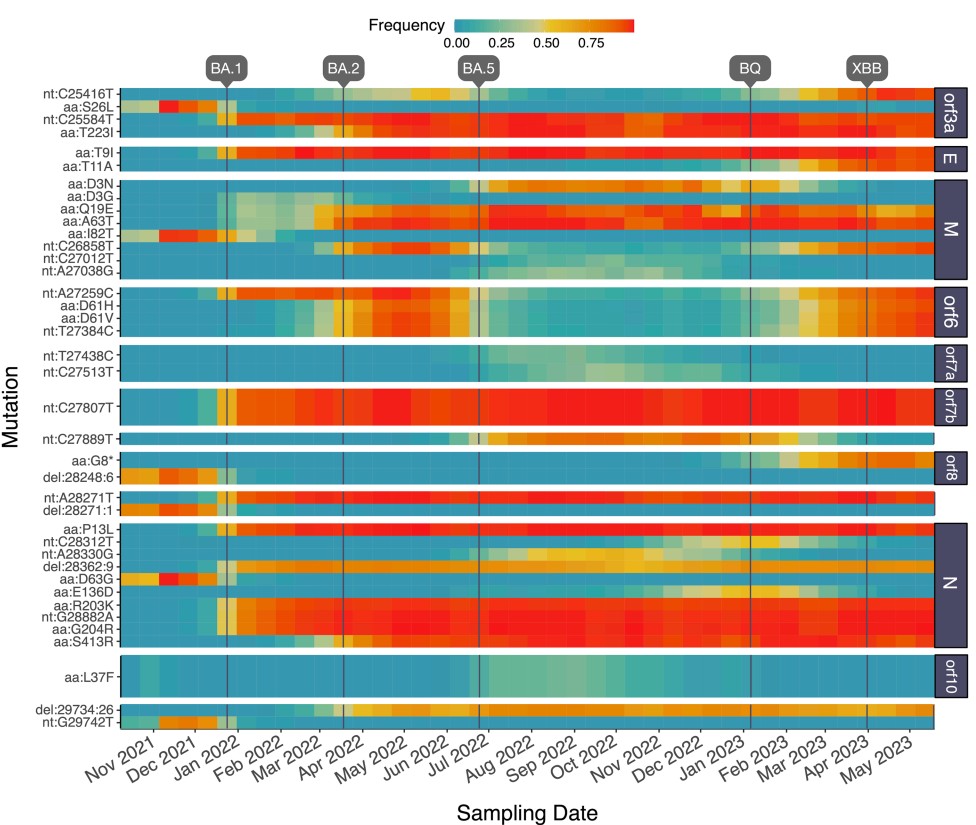

**Fig 6. Heatmaps illustrating the trajectories of SARS-CoV-2 mutations observed at frequencies from 5% to 95% averaged across all samples.** Mutations are ordered top-down by genome coordinate and grouped by open reading frame. Vertical lines indicate dates that major variants (BA.1, BA.2, BA.5, BQ, XBB) became established in Canada, based on CoVaRR-Net SARS-CoV-2 Duotang data. Prefixes 'aa:', 'nt:', 'ins' and 'del' indicate amino acid and silent nucleotide substitutions, insertions and deletions, respectively. The suffix in aa:N460K(G/A) indicates silent nucleotide substitutions subsequent to the amino acid replacement.

Because visualization would not be adequate to detect spatial structure in the distribution of mutation frequencies, we performed a detailed statistical analysis of these data. We calculated the geographic distance between every pair of samples based on the longitude and latitude of the sampling locations (Fig 1). Our expectation is that samples taken from closer locations should tend to have more similar mutation frequencies, i.e., larger dot products, resulting in a negative correlation. Because mutation frequencies were strongly time-dependent, we restricted each correlation to pairs of samples that were taken within the same four week time interval. Fig 8 summarizes the correlation coefficients obtained for each interval, updated by two weeks across the entire range of sample collection dates. We observed negative correlations for a majority ($n = 32$, 80%) of the time intervals. Because these are non-independent observations, we down-sampled these values at a lag of four to eliminate auto-correlation to yield four samples at different offsets. The upper 95% confidence limits for sample means ranged from −0.105 to −0.011 (one-tailed one-sample $t$-tests). These findings indicated that even though spatial structure was not evident in visualizations of genetic variation, there was a statistically significant negative correlation between the genetic similarity and geographic distance between samples.

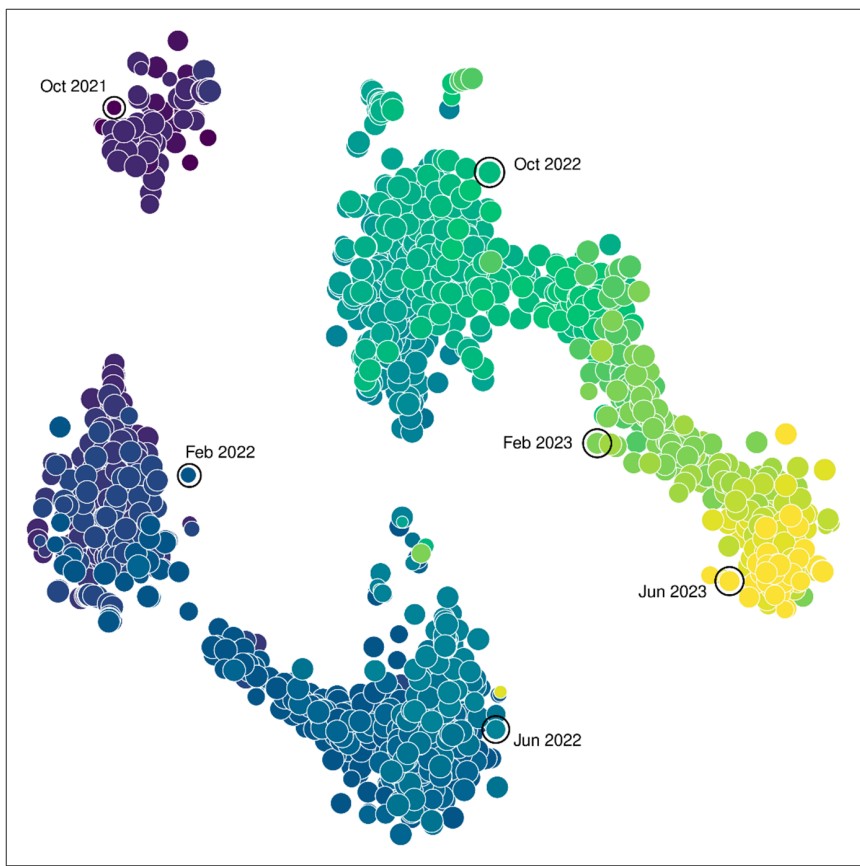

**Fig 7. Visualization of the dot product-based distance matrix relating** $n$ = 1588 **samples using t-stochastic neighbor embedding (t-SNE).** Each point represents a sample, with its area scaled in proportion to coverage (i.e., the number of sites with a minimum depth of 100 reads). Points are coloured using the accessible viridis palette, with the darkest colours corresponding to the earliest sampling months. An arbitrary selection of points (outlined) are also directly labeled with sampling months.

Finally, we conducted a more detailed analysis by randomizing the locations of samples within each month, and then evaluating a genetic clustering test statistic against its null distribution. This test statistic compares the mean genetic similarity (dot product) between samples within a health region to the mean among health regions in the same month of sampling. A test statistic significantly in the upper range of the null distribution would indicate that samples from the same region were more genetically similar than expected by chance. This permutation test revealed spatial clustering patterns that varied by region and month (Fig 9). Overall, the test statistic was significantly greater for 20 out of 70 combinations of region and month with sufficient samples. For instance, a cluster of significantly greater test statistics were associated with the Central East health region for samples between January to May 2022. Samples tended to be more homogeneous across regions in the latter half of the sampling time period (August 2022 to April 2023). However, this may be driven by the reduced rate of sampling outside of the Central West region during this time period (Fig 2), resulting in a loss of power. These results suggest that spatial differences in the genomic variation of SARS-CoV-2 among wastewater samples can be detected at a region-specific level, even at the relatively small scale of a single province.

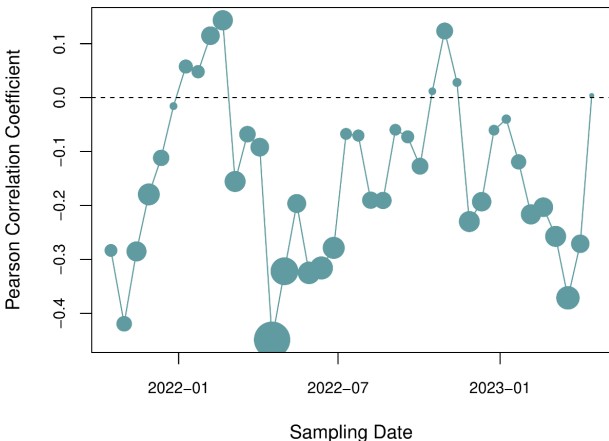

**Fig 8. Pearson correlation coefficient between dot product and geographic distance over time.** Each point represents the Pearson correlation coefficient between the dot product of mutational frequencies and the geographic distance for every pair of samples within a window of four weeks, stepping by two weeks across the range of sampling dates. The area of each point is proportional to the statistical significance of the correlations as quantified by the negative log-transformed P-value.

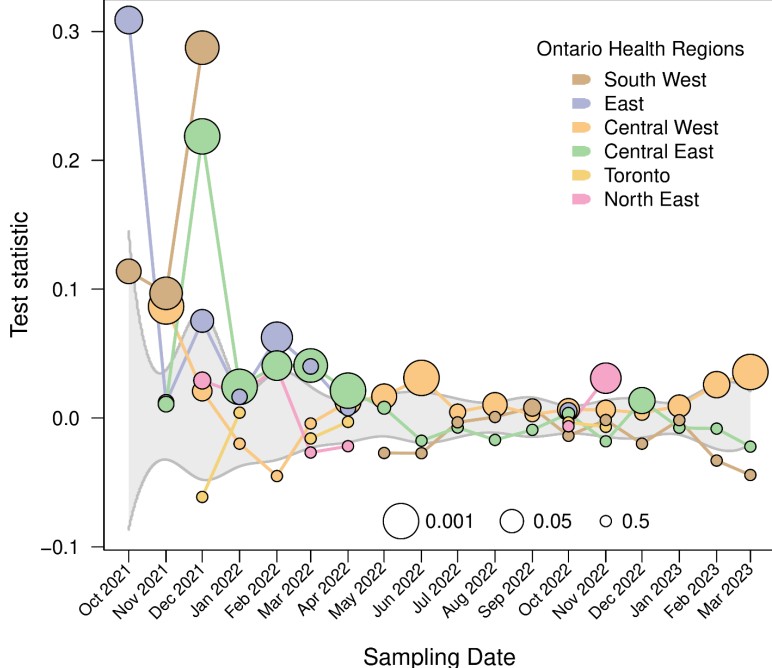

**Fig 9. Permutation test results depicting the difference in mean dot product within and between regions stratified by Ontario health region (OHR).** The gray band summarizes the interquartile range of the null distribution, with quartiles averaged over OHRs and weighted by the number of samples collected each month. Marker sizes represent negative log-transformed proportion of 1,000 randomized replicates exceeding the observed test statistic for a given OHR, to a maximum of 50% (see inset legend).

## Discussion

The targeted sequencing of wastewater samples has been a valuable method for monitoring the evolution and spread of SARS-CoV-2 [16,17,30]. However, these data are also challenging

to interpret due to the rapid diversification of SARS-CoV-2 into thousands of lineages, frequent recombination between lineages, and intermittently low concentrations of viral RNA in wastewater samples [14]. Under the established nomenclature system [28], a SARS-CoV-2 lineage is a group of five or more genome sequences with sufficient coverage and at least one mutation in common. The lineage may be considered to be a basic unit of genomic epidemiology for SARS-CoV-2. In practice, however, monitoring programs have tended to characterize the composition of wastewater samples at a coarser level by grouping related lineages into a single category, which is often labeled with respect to a variant of concern, e.g., 'BA.1.*'. The mixed and fragmented nature of viral RNA in wastewater samples makes it extremely difficult to reconstruct full-length genome sequences that are required to unambiguously assign individual genomes to lineages. Consequently, several 'deconvolution' methods have been developed to indirectly estimate lineage frequencies from these data. These methods have performed fairly well on simulated mixtures of a small number of highly divergent lineages, i.e., Delta (B.1.617.2) and Omicron BA.1 and BA.2 [31]. However, those conditions are not representative of real-world applications of wastewater surveillance where hundreds of closely-related and potentially recombinant lineages may be present in a given sample. In addition, these deconvolution methods generally require prior knowledge on the mutational composition of each lineage, a catalog that must be continuously updated with the ongoing diversification of the virus.

In this study, we have focused on examining the frequencies of mutations rather than lineages to characterize the spatial and temporal distribution of SARS-CoV-2 in Ontario. We observed that the characteristics of mutations detected in wastewater samples were broadly consistent with purifying selection — i.e., lower numbers of non-synonymous substitutions, frameshift-inducing indels — with the exception of an excess of non-synonymous substitutions in the spike protein associated with selective sweeps. Our results demonstrate that analyzing mutations is a promising alternative that can help overcome some of the limitations of lineage-based methods. This is consistent with recent work by Agrawal et al. [32], who found that tracking the frequencies of all SARS-CoV-2 mutations in wastewater samples, rather than focusing on the subset of mutations defining specific lineages, provided more information on how SARS-CoV-2 evolved and spread across different regions of Germany. We observed that, averaged across samples in a given time interval, the trajectories of common mutations were broadly consistent with the introduction and replacement of major lineages. Expanding our analysis to all 241,078 mutations recorded in these samples, we found strong temporal structure, with distinct clusters emerging over the course of the pandemic in Ontario. In addition, our permutation tests indicated that there was statistically significant differentiation between samples from proximal versus distant locations that were collected at similar points in time. However, the fact that we could not readily discern spatial structure among samples using data visualizations (e.g., Fig 7) implies that this result may have limited epidemiological significance.

One of the limitations of this study is that the region-specific sampling rates varied substantially over time. For instance, very few samples were collected from the South West region between January 2022 and April 2022 (Fig 2). This heterogeneity made it difficult to separate the effects of sampling time and geographic location on the mutational composition of samples. In addition, there were several positions in the SARS-CoV-2 genome with particularly low coverage on average (Fig 3). Another potential issue is that we transitioned between three different versions of the ARTIC protocol (V3, V4 and V4.1) over the study period to compensate for effect of mutations on the amplification of specific 'tiles'. It is possible that updating the primer sets may have influenced the number of mutations detected in our data. On the other hand, neglecting to update the protocol would have induced a sampling bias,

as ongoing evolution of the SARS-CoV-2 genome would have reduced the efficiency of PCR amplification over time [33]. It is difficult to assess the impact of these changes on our ability to detect mutations, because the numbers of fixed substitutions and polymorphic sites would have also changed over time. In other words, the effects of sequencing methodologies and the underlying genetic variation are confounded. However, we observed that the distribution of average read depths across tiles was generally consistent over time, with the exception of the first month in our study period (October 2021; S2 Fig). Furthermore, we employed analytical methods (i.e., cosine similarity) that are less sensitive to variation in coverage, as opposed to lineage-based methods that rely on the presence or absence of specific mutations.

A minor limitation of the method that we used to quantify mutation frequencies is that every mutation was recorded independently of any other mutation, even if they occurred in the same read. For example, the occurrence of multiple mutations in codon 371 of the spike protein causes the replacement of the wildtype serine with leucine, but these mutations are recorded separately as S371P and S371F (Fig 6). Moreover, our study was limited by the relatively compact geographic distribution of the wastewater sampling sites, which were mostly collected from the Central West health region (Fig 1), covering about 2,600 square kilometers. This geographic scale may be near the lower limit of detection for spatial structure in wastewater genomic data. On the other hand, other studies have been able to differentiate between locations at a similar geographic scale. For example, Agrawal et al. [34] detected mutations characteristic of the Omicron base lineage B.1.1.529 in wastewater samples weeks before the first clinical case was confirmed by genomic sequencing, and the majority of these mutations were detected in wastewater samples from the Frankfurt Airport instead of the surrounding Frankfurt city area. Subsequent studies also showed that analyzing wastewater in sub-sewer sheds of German metropolitan areas can predict the arrival of a dominant variant [35]. In addition, Rios et al. [36] showed this type of analysis can provide information on the expected spatial scale of SARS-CoV-2 community spread at the resolution of neighborhoods within a city. We also note that additional samples collected and processed by other labs participating in the Ontario Wastewater Sequencing Initiative were available, covering a greater number of wastewater treatment sites in Ontario. However, we decided to restrict our analysis to the samples collected by one lab because there were significant differences in read depth and coverage in sequence data between labs.

## Conclusions

Identifying differences in the mutational composition of SARS-CoV-2 wastewater samples in relation to their location can reveal important information about how viruses spread from their initial point(s) of introduction [17]. If sample compositions within a given time interval were entirely homogeneous with respect to spatial location, a reduction in the number of wastewater surveillance sites in the region could be justified as a cost-effectiveness measure. Our results suggest that such a reduction may be warranted in some settings if the primary objective of the wastewater surveillance program is to capture the dominant trends in SARS-CoV-2 evolution over time, as opposed to tracking the emergence and spread of individual mutations.

## Supporting information

**S1 Fig. Schematic of the workflow from sample collection (blue) to data processing (yellow), analysis and visualization (orange).**
(PDF)

**S2 Fig. Heatmap of average log-transformed read depths for regular intervals of 300 nucleotides across the SARS-CoV-2 genome, stratified by month of sampling.** The average log-transformed read depths are mapped to a colour gradient derived from the accessible 'Plasma' palette in R — a legend is provided on the right-hand side of the plot. The number of samples per month is indicated along the right margin of the plot.
(PDF)

**S3 Fig. Uniform manifold approximation and projection (UMAP) visualization of the dot product-based distance matrix for** $n = 1,587$ **samples.** Similarly to Fig 7, each point represents a sample, with its area scaled in proportion to coverage (number of nucleotide sites with a minimum depth of 100 reads), and coloured by month of sample collection.
(PDF)

**S4 Fig. t-SNE projections of the dot product-based distance matrix with points highlighted by each of the six health regions.** Geographic boundaries of health regions are highlighted in Fig 1. Samples from other regions are represented by small grey points.
(PNG)

## Acknowledgments

We gratefully acknowledge the Ontario Ministry of Environment Conservation (MECP) and the Ontario Clean Water Agency (OCWA) for their support through provision of equipment and technical expertise throughout the Ontario Wastewater Surveillance Initiative. We would also like to thank the contributing researchers, engineers, staff and students from local public health units, academic partners, and wastewater treatment facilities, whose essential roles in wastewater surveillance and sample collection made this work possible.

## Author contributions

**Conceptualization:** Art F. Y. Poon.

**Data curation:** Gopi Gugan, Valeria R. Parreira, Opeyemi U. Lawal, Amber Fedynak, Linkang Zhang, Fozia Rizvi, Melinda Precious, Christopher T. DeGroot, Lawrence Goodridge.

**Formal analysis:** Paula Magbor, William Z. Wang, Abayomi S. Olabode, Art F. Y. Poon.

**Funding acquisition:** Christopher T. DeGroot, Lawrence Goodridge.

**Methodology:** Paula Magbor, William Z. Wang, Gopi Gugan, Devan G. Becker, Art F. Y. Poon.

**Project administration:** Abayomi S. Olabode, Art F. Y. Poon.

**Resources:** Christopher T. DeGroot.

**Software:** Paula Magbor, William Z. Wang, Gopi Gugan, Art F. Y. Poon.

**Supervision:** Art F. Y. Poon.

**Validation:** Paula Magbor, William Z. Wang, Art F. Y. Poon.

**Visualization:** Paula Magbor, William Z. Wang, Art F. Y. Poon.

**Writing – original draft:** Paula Magbor, William Z. Wang, Abayomi S. Olabode, Art F. Y. Poon.

**Writing – review & editing:** Christopher T. DeGroot, Art F. Y. Poon.

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
