## [Decision Letter · Decision Letter 0]

31 Jul 2025

PONE-D-25-23235Spatiotemporal structure of SARS-CoV-2 mutational frequencies in wastewater samples from OntarioPLOS ONE

Dear Dr. Poon,

Thank you for submitting your manuscript to PLOS ONE. After careful consideration, we feel that it has merit but does not fully meet PLOS ONE’s publication criteria as it currently stands. Therefore, we invite you to submit a revised version of the manuscript that addresses the points raised during the review process.

We look forward to receiving your revised manuscript.

Kind regards,

Pablo Colunga-Salas

Academic Editor

PLOS ONE

**Journal Requirements:**

1. When submitting your revision, we need you to address these additional requirements. Please ensure that your manuscript meets PLOS ONE's style requirements, including those for file naming. The PLOS ONE style templates can be found at https://journals.plos.org/plosone/s/file?id=wjVg/PLOSOne_formatting_sample_main_body.pdf and https://journals.plos.org/plosone/s/file?id=ba62/PLOSOne_formatting_sample_title_authors_affiliations.pdf 2. Please update your submission to use the PLOS LaTeX template. The template and more information on our requirements for LaTeX submissions can be found at http://journals.plos.org/plosone/s/latex. 3. Thank you for stating in your Funding Statement: This work was supported by funding from the Ontario Ministry of Environment Conservation and Parks (MECP). Please provide an amended statement that declares *all* the funding or sources of support (whether external or internal to your organization) received during this study, as detailed online in our guide for authors at http://journals.plos.org/plosone/s/submit-now. Please also include the statement “There was no additional external funding received for this study.” in your updated Funding Statement. Please include your amended Funding Statement within your cover letter. We will change the online submission form on your behalf. 4. Please note that your Data Availability Statement is currently missing the repository name. If your manuscript is accepted for publication, you will be asked to provide these details on a very short timeline. We therefore suggest that you provide this information now, though we will not hold up the peer review process if you are unable. 5. Please amend either the abstract on the online submission form (via Edit Submission) or the abstract in the manuscript so that they are identical. 6. We note that Figures 1 and S4 in your submission contain map images which may be copyrighted. All PLOS content is published under the Creative Commons Attribution License (CC BY 4.0), which means that the manuscript, images, and Supporting Information files will be freely available online, and any third party is permitted to access, download, copy, distribute, and use these materials in any way, even commercially, with proper attribution. For these reasons, we cannot publish previously copyrighted maps or satellite images created using proprietary data, such as Google software (Google Maps, Street View, and Earth). For more information, see our copyright guidelines: http://journals.plos.org/plosone/s/licenses-and-copyright. We require you to either present written permission from the copyright holder to publish these figures specifically under the CC BY 4.0 license, or remove the figures from your submission: a. You may seek permission from the original copyright holder of Figures 1 and S4 to publish the content specifically under the CC BY 4.0 license.   We recommend that you contact the original copyright holder with the Content Permission Form (http://journals.plos.org/plosone/s/file?id=7c09/content-permission-form.pdf) and the following text:“I request permission for the open-access journal PLOS ONE to publish XXX under the Creative Commons Attribution License (CCAL) CC BY 4.0 (http://creativecommons.org/licenses/by/4.0/). Please be aware that this license allows unrestricted use and distribution, even commercially, by third parties. Please reply and provide explicit written permission to publish XXX under a CC BY license and complete the attached form.” Please upload the completed Content Permission Form or other proof of granted permissions as an "Other" file with your submission. In the figure caption of the copyrighted figure, please include the following text: “Reprinted from [ref] under a CC BY license, with permission from [name of publisher], original copyright [original copyright year].” b. If you are unable to obtain permission from the original copyright holder to publish these figures under the CC BY 4.0 license or if the copyright holder’s requirements are incompatible with the CC BY 4.0 license, please either i) remove the figure or ii) supply a replacement figure that complies with the CC BY 4.0 license. Please check copyright information on all replacement figures and update the figure caption with source information. If applicable, please specify in the figure caption text when a figure is similar but not identical to the original image and is therefore for illustrative purposes only.The following resources for replacing copyrighted map figures may be helpful: USGS National Map Viewer (public domain): http://viewer.nationalmap.gov/viewer/The Gateway to Astronaut Photography of Earth (public domain): http://eol.jsc.nasa.gov/sseop/clickmap/Maps at the CIA (public domain): https://www.cia.gov/library/publications/the-world-factbook/index.html and https://www.cia.gov/library/publications/cia-maps-publications/index.htmlNASA Earth Observatory (public domain): http://earthobservatory.nasa.gov/Landsat: http://landsat.visibleearth.nasa.gov/USGS EROS (Earth Resources Observatory and Science (EROS) Center) (public domain): http://eros.usgs.gov/#Natural Earth (public domain): http://www.naturalearthdata.com/ 7. We notice that your supplementary figures are included in the manuscript file. Please remove them and upload them with the file type 'Supporting Information'. Please ensure that each Supporting Information file has a legend listed in the manuscript after the references list. 8. If the reviewer comments include a recommendation to cite specific previously published works, please review and evaluate these publications to determine whether they are relevant and should be cited. There is no requirement to cite these works unless the editor has indicated otherwise. 

**Additional Editor Comments:**

Dear authors,

I join the reviewers in thanking them for their clear manuscript and its significant contribution to the study not only of SARS-CoV-2, but also to the epidemiology of emerging diseases. Please review and attend the minor changes, which I believe will make your manuscript a more interesting work.

Reviewers' comments:

Reviewer's Responses to Questions

**Comments to the Author**

1. Is the manuscript technically sound, and do the data support the conclusions?

Reviewer #1: Yes

Reviewer #2: Yes

2. Has the statistical analysis been performed appropriately and rigorously? 

Reviewer #1: Yes

Reviewer #2: Yes

3. Have the authors made all data underlying the findings in their manuscript fully available?

Reviewer #1: Yes

Reviewer #2: Yes

4. Is the manuscript presented in an intelligible fashion and written in standard English?

Reviewer #1: Yes

Reviewer #2: Yes

5. Review Comments to the Author

**Reviewer #1: **Dear authors,

Congratulations to this very nice (and interesting) manuscript. It is clearly written, nicely explained and easy to follow. In the following, please find more concrete comments:

INTRODUCTION

- p. 3, lines 3-11: Suggest to move to methods section as study area sub-section.

METHODS

- p. 4, lines 20-23: May this have an impact on the number of mutations detected?

- p. 5, Figure 1: Please add the number to the legend to show the magnitude of the number of samples, e.g. is a small circle one sample or ten or hundred?

- p. 7, line 1: "This normalization also conferred some robustness to variation in read coverage among samples...", Please discuss this further as it has an impact on your analysis, i.e. deriving spatial or temporal differences when in reality read coverages were different, i.e. not the same mutations were covered.

RESULTS

- General observation: Figures often placed before their call-outs in the text.

DISCUSSION

- p. 20, lines 6-7: Have you considered examining defining mutations for a variant to see whether your approach would be comparable to other data sources in terms of the time component?

- p. 20, lines 14-15: How do you ensure that this is really due to temporal differences in the occurrence rather than differences in coverages?

- p. 20, lines 16-17: Please consider: statistically significant yes, but also practically relevant?

- p. 21, lines 9-10: How did you derive this lower limit of detection for spatial structure, as there are studies showing differences between geographical units on a much smaller spatial scale?

- p. 21, lines 16-17: Your study focused on single mutations instead of lineage-assignment, for which you highlighted the shortcomings, so why bring back this argument in the discussion?

- p. 21, lines 17-21: Please provide an example for residential areas, such as Schmiege, Kraiseburd et al.:

https://doi.org/10.1016/j.scitotenv.2023.165458

- p. 21, lines 21-22: Please provide an example for this statement, such as Rios et al.: https://doi.org/10.1016/j.lanepe.2021.100202

**Reviewer #2: **The authors present a large study of SARS-CoV-2 sequencing of wastewater samples from the Ontario region of Canada. They focus on the nucleotide mutations rather than lineage deconvolution. I found the paper to be interesting, well formatted, and ready for publication. I have no suggestions or questions.

6. PLOS authors have the option to publish the peer review history of their article (what does this mean?). If published, this will include your full peer review and any attached files.

Reviewer #1: No

Reviewer #2: No

---

## [Author Response · Author response to Decision Letter 1]

12 Sep 2025

Please see the attachment ResponseToReviewers.pdf for our detailed response to the reviewer comments.

---

## [Editor Report · Decision Letter 1]

22 Sep 2025

Spatiotemporal structure of SARS-CoV-2 mutational frequencies in wastewater samples from Ontario

PONE-D-25-23235R1

Dear Dr. Poon,

We’re pleased to inform you that your manuscript has been judged scientifically suitable for publication and will be formally accepted for publication once it meets all outstanding technical requirements.

Kind regards,

Pablo Colunga-Salas

Academic Editor

PLOS ONE

Additional Editor Comments (optional):

Dear authors,

I recognize that the work has improved with the reviewers' comments. On the other hand, I am convinced that your manuscript is ready to continue the process and be published.
---

## [Editor Report · Acceptance letter]

PONE-D-25-23235R1

PLOS ONE

Dear Dr. Poon,

I'm pleased to inform you that your manuscript has been deemed suitable for publication in PLOS ONE. Congratulations! Your manuscript is now being handed over to our production team.

Kind regards,

on behalf of

Pablo Colunga-Salas

Academic Editor

PLOS ONE